# HPLC Study of Product Formed in the Reaction of NBD-Derived Fluorescent Probe with Hydrogen Sulfide, Cysteine, *N*-acetylcysteine, and Glutathione

**DOI:** 10.3390/molecules27238305

**Published:** 2022-11-28

**Authors:** Daniel Słowiński, Małgorzata Świerczyńska, Jarosław Romański, Radosław Podsiadły

**Affiliations:** 1Institute of Polymer and Dye Technology, Faculty of Chemistry, Lodz University of Technology, Stefanowskiego 16, 90-537 Lodz, Poland; 2Department of Organic and Applied Chemistry, Faculty of Chemistry, University of Lodz, Tamka 12, 91-403 Lodz, Poland

**Keywords:** biothiols detections, markers, coumarin-derived probe, hydrogen sulfide, fluorescent probe, colorimetric probe

## Abstract

Hydrogen sulfide (H_2_S) and its bioderivatives analogs, such as L-cysteine (L-Cys) and glutathione (GSH), are ubiquitous biological thiols in the physiological and pathological processes of living systems. Their aberrant concentration levels are associated with many diseases. Although several NBD-based fluorescence probes have been developed to detect biological thiols, the HPLC-detection of H_2_S, GSH, L-Cys, and *N*-acetylcysteine-specific products has not been described. Herein, a novel NBD-derived pro-coumarin probe has been synthesized and used to develop a new strategy for the triple mode detection of H_2_S and such thiols as GSH, L-Cys, and NAC. Hydrogen sulfide and those biothiols at physiological pH release fluorescent coumarin from the probe and cause a significant fluorescence enhancement at 473 nm. The appropriate NBD-derived product for H_2_S, L-Cys, GSH, and NAC has a different color and retention time that allows distinguishing these biological thiols meaning the probe has a great possibility in the biological application. Fluorescent imaging combined with colorimetric and HPLC detection of H_2_S/biothiol-specific product(s) brings a potential tool for confirming the presence of biological thiols and determining concentrations in various aqueous biological samples.

## 1. Introduction

Hydrogen sulfide (H_2_S), L-cysteine (L-Cys), homocysteine (Hcy), and glutathione (GSH) are the most common small molecular biological thiols [1]. They are closely related and have essential distinct biological and pharmacological roles in many processes [2]. Hydrogen sulfide plays a vital and physiological function in the human body [3]. That small molecule, although commonly recognized as toxic gas and environmental hazard, is now accepted as a critical, endogenously produced signaling mediator, similar to carbon monoxide (CO) and nitric oxide (NO) [4,5]. Hydrogen sulfide rapidly travels through cell membranes without additional transporters and does not have one single pathway- it affects multiple cellular effectors in a cell or tissue-dependent manner [6]. In humans, the concentration of H_2_S in the body can differ according to age, the kind of tissues, and measuring methods. For example, the level of H_2_S in the peripheral blood is generally 30–100 μM, while the H_2_S level in the mammalian brain is 50–160 μmol/L [7,8]. According to current research, cells produce H_2_S via the enzymatic activity of cystathionine-γ-lyase (CSE) and cystathionine-β-synthase (CBS) [9]. Besides this synthetic pathway, H_2_S is also produced by the activity combination of the cysteine aminotransferase (CAT) and 3-mercaptopiruvate sulfurtransferase (MST) [10], and can also be made endogenously through nonenzymatic pathways. The glycolytic pathway can reduce thiosulfate, thiocysteine, and other sulfur-containing molecules in the blood to H_2_S [11].

Hydrogen sulfide is related to many physiological processes, including vasodilation, neurotransmission, angiogenesis, inhibition of insulin signaling, antioxidation, and regulation of blood vessel tone. It can also effectively neutralize reactive oxygen and nitrogen species (ROS and RNS) [12,13,14,15,16,17,18]. In recent years, treatment of myocardial ischemia injury with H_2_S has emerged as a novel and promising strategy to protect cardiac structure and function. It was reported that H_2_S could directly increase the production of reduced glutathione (GSH) to affect cytoprotective effects against ROS-mediated damage [19]. However, abnormal levels of H_2_S are associated with human health and many diseases [20], such as diabetes [21,22], arterial and pulmonary hypertension [23], gastric mucosal damage [24], liver cirrhosis [25,26], ischemia [27], and Alzheimer’s disease [28,29].

In the last decade, some analytical methods (gas chromatography, methylene blue assay, sulfur-selective electrodes) were developed for H_2_S detection, but they are unsuitable for in situ analysis [30,31]. In contrast to traditional methods, techniques based on fluorescent probes could maintain better accuracy and efficiency, higher selectivity and sensitivity, and noninvasive real-time imaging [32]. Based on nucleophilic addition [33], reduction of azide or nitro group to amine [34], copper sulfide precipitation reaction [35], and thiolysis of ethers [36,37,38], a wide range of reaction-based H_2_S-specific probes have been developed and applied in living biological systems. In the H_2_S-induced thiolysis reaction, the design strategy of the H_2_S fluorescent probes is based on hydrosulfide anion (HS^−^) selectively attacking a single electrophilic position on the probe scaffold, which is followed by the removal of electrophilic functionalities to generate fluorescent signals. The fluorescence of these probes is typically protected with an incorporated strong electron-withdrawing group such as naphthoquinone [39] or nitrobenzofuran (NBD) that can be cleaved in the presence of H_2_S [40]. Probes derived from NBD moiety are widely applied in biological applications due to their small size and good photophysical properties in aqueous systems [41].

Nevertheless, most reported probes suffered from limitations such as narrow Stokes shift (<60 nm) or a long response time. Additionally, many probes cannot distinguish H_2_S from other biothiols because they all contain a sulfhydryl group with similar nucleophilic properties [42]. Therefore, searching for an effective tool capable of simultaneously monitoring the level of H_2_S in vivo, along with the possibility of distinguishing selected thiols, is a significant challenge. Although several NBD-based fluorescence probes have been developed to detect biothiols, the HPLC-detection of NBD-derived specific products for H_2_S, GSH, Cys, or NAC has not been characterized.

In this report, we describe the design of a novel fluorescent probe (NBD-O-CmCH_2_OH) consisting of a fluorophore of 7-hydroxy-4-(hydroxylmethyl)-2H-chromen-2-one (CmCH_2_OH) connected via ether linkage with 7-nitrobenz-2-oxa-1,3-diazole (NBD) moiety. Treatment of this probe with H_2_S or biothiols rapidly cleaved the C-O bond to afford coumarin fluorophore and the diagnostic marker products of H_2_S, Cys, GSH, and NAC, easily detected by HPLC (Figure 1). Moreover, these specific products have different spectral characteristics, resulting in different solutions’ colors. Therefore, combining non-invasive fluorescence/colorimetric monitoring with HPLC analyses to detect H_2_S/biothiols-specific products will provide more detailed information on the species identity.

## 2. Results and Discussion

### 2.1. Synthesis of the Probe and H_2_S/Biothiols-Specific Product

First, we designed and synthesized a pro-coumarin NBD-derived probe to develop the strategy for triple mode detection of H_2_S and biothiols (GSH, Cys, NAC). Previously reported NBD-coumarin probes [43,44,45,46,47] were used in aqueous solutions with the addition of a large amount (above 30% *v/v*) of organic co-solvent. To improve the probe’s solubility, we choose fluorophore 7-hydroxy-4-(hydroxylmethyl)-2H-chromen-2-one (CmCH_2_OH) with an additional hydroxyl group. The probe was obtained using the three-step route with a good yield (Figure 2). 4-Chloromethyl-7-hydroxycoumarin CmCH_2_Cl was synthesized from commercially available ethyl (2-chloroaceto)acetate and resorcinol [48]. Next, the product was converted into fluorophore CmCH_2_OH [49]. Finally, the probe NBD-O-CmCH_2_OH in 69% yield was prepared by coupling CmCH_2_OH with NBD-Cl in the presence of triethylamine.

7-nitrobenzo[c][1,2,5]oxadiazole-4-thiol (NBD-SH) was synthesized from NBD-Cl and sodium sulfide. Biothiols-specific products, 7-nitrobenzo[c][1,2,5]oxadiazole-4-glutathione (NBD-GSH), 7-nitrobenzo[c][1,2,5]oxadiazole-4-cysteine (NBD-Cys), *N*-acetyl-S-(7-nitrobenzo[c][1,2,5]oxadiazol-4-yl)cysteine (NBD-NAC) were obtained by mixing NBD-Cl with an excess of proper biothiol (Figure 3) at room temperature in the presence of triethylamine. The structure of all synthesized products was confirmed by ^1^H NMR, ^13^C NMR, and HRMS spectroscopy (see Appendix A). The absorbance and emission profiles of CmCH_2_OH, probe, and NBD-Cl are illustrated in Appendix A.

### 2.2. Spectral Response of NBD-O-CmCH_2_OH toward Na_2_S and Biothiols

Several coumarin-based compounds containing the NBD moiety as the sensing group have been used as fluorescent probes for the thiol species (GSH, Cys, Hcy, H_2_S) [43,44,45,46,47]. In our research, we also decided to test the reactivity of the probe toward the cysteine pro-drug, *N*-acetylcysteine (NAC). Our results in Figure 1 and Appendix A confirmed that studied sulfhydryl compounds could release the fluorescent coumarin from NBD-O-CmCH_2_OH. The fluorescent signal reaches a plateau within 10 min in phosphate buffer (PB buffer) containing 10% acetonitrile (MeCN) (*v/v*). The time-dependent fluorescence at 473 nm was used to determine reaction kinetics. The pseudo-first-order rates (k_obs_) were found by fitting the data with a single exponential function (Appendix A). The determined kinetic parameters (Table 1) also show that the tested probe reacts with thiols at similar rate constants. These results indicated that the NBD-based probe could respond efficiently with micromolar-tested biothiols under physiological conditions.

Next, we investigated the colorimetric responses of NBD-O-CmCH_2_OH in the absence and presence of thiol species. The electronic absorption spectra recorded after the bolus addition of sulfhydryl compounds are presented in Figure 2.

As shown in Figure 2, the probe (dark line) displayed an absorption peak at 375 nm. A common feature was that an absorption peak at 324 nm was observed after treatment of NBD-O-CmCH_2_OH with Na_2_S and biothiols. Moreover, the reaction between NBD-O-CmCH_2_OH and thiol species led to a new absorption band(s) located at the wavelength listed in Table 2. Based on the previous articles [44,46], the individual absorption bands were assigned to the specific products formed during the reaction between thiols and probe. Reaction with: (i) Na_2_S gave NBD-SH, (ii) L-Cys formed NBD-Cys adduct exhibiting the absorption of 475 nm consistent with the formation of amino-bound NBD, (iii) GSH generated NBD-GSH compound with the absorption of 418 nm consistent with the formation of sulfur-bound NBD and (iv) NAC formed NBD-NAC adduct with the absorption bands centered at 424 and 543 nm.

The results show that the colorimetric analysis of single biothiols can give satisfactory results. NBD-coumarin probe can differentiate H_2_S from GSH, L-Cys, and NAC, or sulfur species (S_2_O_4_^2−^, SO_3_^2−^, HSO_3_^−^), when only single components were analyzed (Figure 3). However, in a mixture of thiol species, their colorimetric identification may lead to wrong conclusions, especially when we intend to study in this way the influence of NAC on the level of GSH, H_2_S, or L-Cys. In addition, we compared the change in fluorescence intensity during the reaction of NBD-Cl with analytes and the probe NBD-O-CmCH_2_OH with analytes. (Appendix A).

### 2.3. HPLC and Fluorescence Measurements

Over the last decade, it has been emphasized that fluorescent identification of analyte formed in cells is practically impossible without employing HPLC- or LC-MS-based methods to detect analyte-specific product(s) [39]. Therefore, we have developed HPLC-based methods to detect the specific reaction products of the NBD-O-CmCH_2_OH probe with the tested thiols. Each of these reactions produces a fluorescent coumarin CmCH_2_OH (retention time 1.95 min) and NBD-specific adducts: NBD-SH (2.99 min), NBD-GSH (2.50 min), NBD-NAC (3.25 min), and NBD-Cys (3.45 min) (Figure 4A). A comparison of the retention time of the authentic standards, which were obtained by reacting NBD-Cl with an appropriate biothiol, confirms that these NBD adducts are the only products found in the reaction mixture (Figure 4B). NBD-SH, NBD-GSH, and NBD-NAC are S-bound NBD adducts. The S-bound NBD product of L-Cys is unstable and, in the presence of an excess of L-Cys, can quickly change into the corresponding N-bound NBD product via a unique nucleophilic aromatic substitution-intramolecular Smiles rearrangement, which we show later in this work (Figure 4). Using our probe in conjunction with the HPLC technique, we can detect the presence of specific biothiol in a mixture.

Next, we examined the stoichiometry of the probe reaction with the tested compounds. We used two methods to determine the reaction’s stoichiometry between the probe and tested thiols. Initially, HPLC titration of NBD-O-CmCH_2_OH with increasing concentration of Na_2_S and studied biothiols were performed. Chromatograms (Figure 5A) show the slow disappearance of the probe (NBD-O-CmCH_2_OH, r_t_ = 4.15 min) and the formation of a fluorescence product with a retention time of 1.95 min. A comparison of the retention time of the authentic standard of 7-hydroxy-4-(hydroxylmethyl)-coumarin (CmCH_2_OH) confirms that this coumarin is one of the products found in the reaction mixture. Our stoichiometric analysis (Figure 5B) showed that three equivalents of Na_2_S completely consumed the NBD-O-CmCH_2_OH probe.

In addition, the concentration-dependent fluorescence measurements of NBD-O-CmCH_2_OH toward Na_2_S, L-Cys, GSH, and NAC were performed. Recorded emission spectra showed gradual enhancement in fluorescence intensity at 473 nm with an increased concentration of biothiols (Figure 6). Using these fluorescence measurements, the detection limits (LOD = 3.3σ/S) of NBD-O-CmCH_2_OH for H_2_S, L-Cys, GSH, and NAC were calculated on 140 nM, 26 nM, 60 nM, and 32 nM, respectively (Table 1). The dependence of the fluorescence intensity on the added thiol concentration shows that the emission of the probe reaches a maximum for various concentrations of Na_2_S and biothiols. To get the emission plateau, four equivalents of Na_2_S and one equivalent of biothiol are needed.

We also recorded HPLC chromatograms during the reaction of the NBD-O-CmCH_2_OH probe with various biothiols (Figure 7). The stoichiometric analysis showed that all tested biothiols entirely consumed the NBD-derived probe. The maximum yield of CmCH_2_OH was achieved when the probe was reacted with a slight excess of the biothiols consistent with the 1:1 or 1:2 stoichiometry. The probe reacting with 0.5 equivalent of L-Cys gives a fluorescent coumarin and a new product with a retention time of 4.5 min, which can be assigned to NBD-S-Cys. The stoichiometric amount of L-Cys caused a significant disappearance of this product and the formation of a proper NBD-Cys adduct with a retention time of 3.5 min. The addition of two equivalents of L-Cys caused the complete conversion of NBD-S-Cys to NBD-Cys (Figure 7A). These studies confirm that the S-bound NBD adduct of L-Cys can undergo intramolecular rearrangement under these conditions. Moreover, we do not observe any intermediate products during the probe’s reaction with GSH or NAC (Figure 7B,C). These thiols could not induce this intramolecular rearrangement reaction due to the lack of a proximal NH_2_ or blocked amino group. For better visualization of the formation of specific NBD adducts, we also recorded the HPLC chromatogram at 420 nm (Appendix A).

## 3. Materials and Methods

### 3.1. Materials and Instruments

All reagents for synthesis were obtained from commercial suppliers and used without purification. Distilled water was used throughout all experiments. ^1^H NMR and ^13^C-NMR spectra were recorded with a Bruker Avance III 600 using as solvent DMSO-*d_6_* or CD_3_OD. High-resolution mass spectra were taken on using Synapt G2-Si mass spectrometer equipped with an electrospray ionization (ESI) source and time of flight (TOF) analyzer. UV-vis absorption spectra were recorded on a Jasco V-670 spectrophotometer. Fluorescence spectra were recorded using FLS-920 (Edinburgh Instruments, UK) with excitation and emission slit widths of 1 nm. HPLC chromatograms were obtained using UFLC Shimadzu equipped with UV-Vis absorption and fluorescence detector. Analyses were done using a Kinetex C18 column (Phenomenex 100 mm × 46 mm, 2.6 µm) equilibrated with 10% MeCN in water containing 0.1% trifluoroacetic acid (TFA).

### 3.2. Synthesis

The synthetic pathway starting from resorcinol toward NBD-O-CmCH_2_OH is depicted in Figure 2. The 4-chloromethyl-7-hydroxycoumarin (CmCH_2_Cl) and 7-hydroxy-4-(hydroxylmethyl)-2H-chromen-2-one (CmCH_2_OH) were prepared according to a slightly modified procedure described elsewhere [48,49]. The following derivatives of NBD, such as NBD-SH, NBD-Cys, NBD-GSH, and NBD-NAC, were obtained by the straightforward synthesis depicted in Figure 3.

#### 3.2.1. Synthesis of 4-Chloromethyl-7-Hydroxycoumarin (CmCH_2_Cl)

A solution of resorcinol (3.30 g, 30 mmol) in ethyl 4-chloroacetoacetate (5.52 mL, 40 mmol) was added dropwise to cooled, concentrated sulfuric acid (20 mL) at 5 °C. Then the reaction mixture was stirred at room temperature for 2 h. The mixture was poured into ice water, and the precipitate was collected by suction filtration and washed with cold water, dried to afford pure CmCH_2_Cl as a colorless solid (5.14 g, 82%), R_f_ = 0.89 (DCM:MeOH 17:3, *v/v*), m.p. 187–188 °C (186–188 °C [48]). ^1^H NMR (DMSO-*d_6_*, 600 MHz) δ (ppm): 4.96 (2H, s); 6.42 (1H, s); 6.76 (1H, d, J = 2.4 Hz); 6.85 (1H, dd, J_1_ = 6.6 Hz, J_2_ = 2.4 Hz); 7.68 (1H, d, J = 8.4 Hz); 10.64 (1H, s); ^13^C NMR (DMSO-*d_6_*, 151 MHz) δ (ppm): 41.8; 103.0; 109.8; 111.5; 113.6; 127.0; 151.4; 155.8; 160.6; 161.9; HRMS [M-H]^-^ for C_10_H_6_ClO_3_ 209.0005, found 209.0010.

#### 3.2.2. Synthesis of 7-Hydroxy-4-(Hydroxylmethyl)-2H-Chromen-2-One (CmCH_2_OH)

CmCH_2_Cl (1.5 g, 7 mmol) was added to water (100 mL). The reaction mixture was refluxed for 48 h, then filtered while hot and cooled to room temperature to yield a white precipitate. The product was filtered, washed with a large amount of cold water, and chromatographed on silica gel (DCM:MeOH 17:3, *v/v*) to give CmCH_2_OH as a white solid (R_f_ = 0.78, 1.2 g, 89%), m.p. 213–217 °C (216–218 °C [49]). ^1^H NMR (DMSO-*d_6_*, 600 MHz) δ (ppm): 4.71 (2H, s); 5.57 (1H, s); 6.25 (1H, s); 6.73 (1H, d, J = 1.8 Hz); 6.77 (1H, dd, J_1_ = 6.0 Hz, J_2_ = 2.4 Hz); 7.68 (1H, d, J = 8.4 Hz); ^13^C NMR (DMSO-*d_6_*, 151 MHz) δ (ppm): 59.5; 102.8; 107.1; 110.1; 113.3; 125.9; 155.4; 157.2; 161.1; 161.5; HRMS [M-H]^-^ for C_10_H_7_O_4_ 191.0344, found 191.0343.

#### 3.2.3. Synthesis of 7-Nitrobenzofurazan Ether-4-Hydroxylmethylcoumarin (NBD-O-CmCH_2_OH)

CmCH_2_OH (0.15g, 0.78 mmol) and triethylamine (0.22 mL, 1.6 mmol) were dissolved in anhydrous ethanol (30 mL), and next, NBD-Cl (0.16 g, 0.8 mmol) was added in one portion to the flask, and the resulting mixture was stirred at room temperature for 8 h. After removal of the solvent, the residue was chromatographed on silica gel using DCM:MeOH (17:3, *v/v*) to afford pure NBD-O-CmCH_2_OH as a yellow solid (R_f_ = 0.85 (DCM:MeOH 17:3, *v/v*), 0.19 g, 69%), m.p. 215–217 °C. ^1^H NMR (DMSO-*d_6_*, 600 MHz) δ (ppm): 4.81 (2H, s); 5.73 (1H, s); 6.51 (1H, s); 7.02 (1H, d, J = 8.4 Hz); 7.42 (1H, dd, J_1_ = 6.6 Hz, J_2_ = 2.4 Hz); 7.91 (1H, d, J = 9.0 Hz); 8.68 (1H, d, J = 8.4 Hz); ^13^C NMR (DMSO-*d_6_*, 151 MHz) δ (ppm): 59.6; 109.3; 110.9; 112.3; 116.2; 117.1; 127.2; 131.7; 135.6; 144.9; 145.9; 152.2; 154.8; 156.0; 156.5; 160.2; HRMS [M + H]^+^ for C_16_H_10_N_3_O_7_ = 356.0519, found 356.0528.

#### 3.2.4. Synthesis of 7-Nitrobenzo[c][1,2,5]Oxadiazole-4-Thiol (NBD-SH)

4-Chloro-7-nitrobenzo[c][1,2,5]oxadiazole (NBD-Cl, 0.1 g, 0.5 mmol) was dissolved in 5 mL of MeOH. Next, anhydrous Na_2_S (0.078 g, 1 mmol) was dissolved in 5 mL of MeOH and slowly added to the solution of NBD-Cl. The reaction mixture changed from yellow to deep purple and was stirred at room temperature for 30 min. Next, the MeOH was removed under a vacuum to afford the desired product as dark purple powder. The final product was purified by chromatography on SiO_2_ using DCM:MeOH (17:3, *v/v*) (0.079 g, 82%). ^1^H NMR (DMSO-*d_6_*, 600 MHz) δ (ppm): 5.75 (1H, d, J = 9.6 Hz); 8.25 (1H, d, J = 10.2 Hz); ^13^C NMR (DMSO-*d_6_*, 151 MHz) δ (ppm): 111.5; 115.6; 123.1; 138.3; 147.2; 148.6; HRMS [M-H]^-^ for C_6_H_2_N_3_O_3_S: 195.9817, found 195.9820.

#### 3.2.5. Synthesis of 4-Nitrobenz-2-Oxa-1,3-Diazole (NBD) Derived Compounds: NBD-Cys, NBD-GSH, NBD-NAC

NBD-Cl (0.15 g, 0.7 mmol) and triethylamine (0.146 mL, 1.05 mmol) were added to EtOH (20 mL), and then a solution of proper biothiol (1.4 mmol) in 5 mL of water was added to the flask. The resulting mixture was stirred for 3h at room temperature. The resulting precipitate was filtered and washed with cold ethanol to give a crude product which was then purified by chromatography on SiO_2_ using DCM:MeOH (17:3, *v/v*).

NBD-Cys (cysteine derivative): orange solid, yield 85%; ^1^H NMR (CD_3_OD, 600 MHz) δ (ppm): 3.66 (1H, br. s); 4.55 (2H, br. s); 6.42 (1H, d, J = 9.0 Hz); 8.52 (1H, d, J = 8.4 Hz); HRMS [M-H]^-^ for C_9_H_7_N_4_O_5_S: 283.0137 found 283.0141.

NBD-GSH (glutathione derivative): yellow solid, yield 68%; ^1^H NMR (DMSO-*d_6_*, 600 MHz) δ (ppm): 1.90–1.98 (2H, m); 2.36 (2H, t, J = 7.2 Hz); 3.38 (2H, t, J = 6.0 Hz); 3.53–3.57 (2H, m); 3.76–3.80 (2H, m); 4.72–4.75 (2H, m); 7.64 (1H, d, J = 8.4 Hz); 8.59 (1H, d, J = 7.8 Hz); 8,75; 8,81 (2H, d-like); ^13^C NMR (DMSO-*d_6_*, 151 MHz) δ (ppm): 27.1; 31.8; 33.5; 41.8; 51.6; 53.5; 123.1; 132.8; 132.9; 139.6; 143.1; 149.6; 170.2; 171.1; 171.3; 173.8; HRMS [M-H]^-^ for C_16_H_17_N_6_O_9_S: 469.0778, found 469.0785.

NBD-NAC (*N*-acetylcysteine derivative): brown solid, yield 73%; ^1^H NMR (DMSO-*d_6_*, 600 MHz) δ (ppm): 1.87 (3H, s); 2.53 (2H, d-like); 4.66 (1H, d-like); 7.60 (1H, d, J = 6.6 Hz); 8.53 (1H, d, J = 6.6 Hz); 8.60 (1H, d, J = 6.6 Hz); ^13^C NMR (DMSO-*d_6_*, 151 MHz) δ (ppm): 22.8; 33.0; 52.1; 123.3; 132.8; 133.0; 139.3; 143.1; 149.7; 170.1; 171.7; HRMS [M-H]^-^ for C_11_H_9_N_4_O_6_S: 325.0243, found 325.0244.

### 3.3. UV-Vis, Fluorescent, and HPLC Analysis

All spectroscopic measurements were performed in phosphate buffer (0.1 M, pH 7.4) containing 10% MeCN at room temperature. Probe NBD-O-CmCH_2_OH and fluorescent standard CmCH_2_OH were dissolved in MeCN to prepare 1 mM solutions. Before each experiment probe was diluted with phosphate buffer (0.1 M, pH 7.4 with 10% MeCN) to afford the final concentration. Sodium sulfide (Na_2_S) was selected as a H_2_S donor. The stock solution of Na_2_S (1 mM) in PB buffer (0.1 M, pH 7.4) was always prepared before experiments. To evaluate hydrogen sulfide concentration in phosphate buffer, we used Ellman’s reagent, i.e., 5,5′-dithiobis(2-nitrobenzoic acid (DTNB), and a detailed procedure is described in our previous paper [39]. Aqueous stock solutions (1 mM) of an amino acid (Cys, GSH, NAC) and sodium salts (SO_4_^2−^, S_2_O_4_^2−^, SO_3_^2−^, HSO_3_^−^) were prepared using deionized water. All measurements were performed in a 3.5 mL quartz cuvette with a 2 mL solution. The reaction mixture was shaken uniformly at room temperature before recording spectra. For fluorescence measurement, the excitation wavelength was set at 320 nm. The adjustment slit was 1 nm/1 nm for each fluorescence measurement. The standards and products formed in the reaction of NBD-O-CmCH_2_OH with H_2_S and tested biothiols were eluted by an increase of MeCN concentration from 10 to 100% over 12 min at the flow rate of 1.5 mL/min. The HPLC trace of NBD-O-CmCH_2_OH and CmCH_2_OH formed in reaction with H_2_S was detected by monitoring the absorption at 330 nm. HPLC traces of NBD-SH and NBD-Cys, NBD-GSH, and NBD-NAC were detected by monitoring absorption at 530 nm and 420 nm, respectively.

## 4. Conclusions

In the current investigations, a coumarin analog fluorescent probe has been applied to indicate the hydrogen sulfide and the naturally occurring biothiols (cysteine, *N*-acetylcysteine, and glutathione). The design and synthesis were described, and the NBD-derived probe was reacted with H_2_S, L-Cys, NAC, and GSH to show similar rate constants. The differences in the stoichiometry of the probe response to H_2_S, GSH, L-Cys, and NAC are reflected in the detection limits. The probe’s ether bond cleavage by the corresponding biothiol releases the coumarin fluorophore and produces thiol-NBD adducts. The appropriate NBD-derived products with H_2_S, Cys, NAC, and GSH possess various colors and retention times that distinguish those biological thiols. We also confirmed that intramolecular rearrangement is only possible for the adduct formed during the reaction of the probe NBD-O-CmCH_2_OH with L-Cys. The resulting NBD-S-Cys cleavage product can rapidly transform to create a new NBD-Cys adduct only when excess L-Cysteine is used.

## Data Availability

Not applicable.

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
