# Peer review of "HPLC Study of Product Formed in the Reaction of NBD-Derived Fluorescent Probe with Hydrogen Sulfide, Cysteine, *N*-acetylcysteine, and Glutathione"

_molecules, 2022, doi:10.3390/molecules27238305_

Round 1
Reviewer 1 Report
The work describes the probe in which NBD and coumarine are combined. This is a very detailed study with careful analysis of the products formed in the reaction with SH-containing substrates. It is indeed rare that the products of reactive probes are investigated and the structures are clarified. The authors determined LOD, HPLC yields of the products, measured fluorescence spectra. This work definitely deserves publication. I have one minor comment. NBD-Chloride reacts also with all investigated substrates. However, the spectra of the products from the reaction NDB-Cl+ analytes and NDB-Coumarine + analytes were not compared. Please insert NMR and fluorescence spectra from both reactions. The structure of the GSH adduct should be also discussed, is GSH connected to NDB thought NH or SH?
Author Response
Comments and answers for reviewers.
Reviewer 1
The work describes the probe in which NBD and coumarine are combined. This is a very detailed study with careful analysis of the products formed in the reaction with SH-containing substrates. It is indeed rare that the products of reactive probes are investigated and the structures are clarified. The authors determined LOD, HPLC yields of the products, measured fluorescence spectra. This work definitely deserves publication. I have one minor comment. NBD-Chloride reacts also with all investigated substrates.
However, the spectra of the products from the reaction NDB-Cl+ analytes and NDB-Coumarine + analytes were not compared. Please insert NMR and fluorescence spectra from both reactions.
According to the Reviewer’s recommendation, we added fluorescence spectra recorded during reactions between NBD-Cl / probe with the tested sulfur species compounds (Figure S4 in Supplementary Materials). We thank for this suggestion.
The structure of the GSH adduct should be also discussed, is GSH connected to NDB thought NH or SH?
According to Scheme 4 and the information in the article, NBD-GSH is an S-bound adduct incapable of intramolecular rearrangement to form an N-bound adduct due to the lack of a proximal amino group. (see page 7 and Scheme 4)
Reviewer 2 Report
The authors described a new method concerning HPLC study of product formed in the reaction of NBD-derived fluorescent probe with H₂S, Cys, NAC, and GSH. The topic is interesting. The article is well written and organized. However, many revisions are still required.
1- Title should be written with full names [ no abbreviations should be used ]
2- NAC-specific products NAC full name should be stated in the first mentioning.
3- The following articles should be cited
https://pubs.rsc.org/en/content/articlelanding/2014/OB/C3OB41870G
https://www.sciencedirect.com/science/article/pii/S2090123220301028
https://www.proquest.com/openview/07dad5b445be3ce7587cb272664aa2cb/1?pq-origsite=gscholar&cbl=18750
4- Line 195, MeCN full name should be stated
5- I can not find the real application of the method on biological samples ? did the authors tried it ?
6- Future research plan and study limitation should be provided
7- Comparative table for the new method with the old methods in terms of LOD,LOQ, interferences , applications, linearity ranges, merits and demerits should be added in the discussion
8- Novelty statement should be described in details in the abstract with real application of the method
9- Abbreviation list should be provided
Best wishes
Author Response
Reviewer 2
The authors described a new method concerning HPLC study of product formed in the reaction of NBD-derived fluorescent probe with H₂S, Cys, NAC, and GSH. The topic is interesting. The article is well written and organized. However, many revisions are still required.
- Title should be written with full names [ no abbreviations should be used ]
According to the Reviewer’s recommendation, we changed the title of the manuscript. We have replaced the abbreviations with the full names of the compounds.
- NAC-specific products NAC full name should be stated in the first mentioning.
We thank for this suggestion. The full names of all specific products are included in the article in the first mentioning. In addition, we have included a list of abbreviations in the article.
3- The following articles should be cited
https://pubs.rsc.org/en/content/articlelanding/2014/OB/C3OB41870G
https://www.sciencedirect.com/science/article/pii/S2090123220301028
https://www.proquest.com/openview/07dad5b445be3ce7587cb272664aa2cb/1?pq-origsite=gscholar&cbl=18750
According to the Reviewer’s recommendation, we have added the suggested articles.
- Line 195, MeCN full name should be stated
The abbreviation of MeCN is explained on page 4 of the article on line 134. We thank for this suggestion.
5- I can not find the real application of the method on biological samples? did the authors tried it ?
In this article, we focused only on developing a novel probe and technique for detecting the presence of specific biological thiols (such as cysteine, glutathione, and N-acetylcysteine). We have performed all the necessary spectroscopic measurements, which can be a base for future biological research. We have not yet started any testing on biological samples at this time.
6- Future research plan and study limitation should be provided.
This article focuses on the chemical reactivity of NBD-derived probe with H₂S, Cys, NAC, and GSH. Our primary goal was to show how to detect selected biothiols using a simple HPLC system. We plan to use our probe to detect thiols-species using fluorescence-based detection in a cellular system. However, this will require the synthesis of more probe and the establishment a cell model or H2S-generating enzyme system. The lack of these results means that we do not want to speculate on our approach's advantages and disadvantages in detecting specific reaction products of NBD with h2,s, and GSH.
7- Comparative table for the new method with the old methods in terms of LOD,LOQ, interferences , applications, linearity ranges, merits and demerits should be added in the discussion
The novelty in this article is the design of a modern probe and an innovative approach to detecting selected biothiols using the characteristic retention times of products formed between the probe and biothiols. The detection limit was determined in a traditional way using fluorescence measurements. The linearity ranges measured on different instruments are shown in Figures 5,6,7 and S3. In addition, we included a table comparing our developed probe NBD-O-CmCH2OH with other previously reported fluorescent probes (Table 1 in Supplementary Materials).
8- Novelty statement should be described in details in the abstract with real application of the method
According to the Reviewer’s recommendation, we mentioned in the abstract the potential use of a probe and the application of the method using HPLC and fluorescence technique. We thank for this suggestion.
9- Abbreviation list should be provided
According to the Reviewer’s recommendation, we have included a list of abbreviations at the end of the article. We thank for this suggestion.
Round 2
Reviewer 2 Report
thanks for your fruitful responses. the paper could be published in the current form